# Variabilities and contentions in anesthesiologists' perspectives on Japanese perianesthesia nurses: A qualitative study

Mikiko Tamai [1,2]*, Shogo Kojima[3], Yasuko Baba[1], Kiyoyasu Kurahashi[4]

**1** Department of Anesthesiology, International University of Health and Welfare Mita Hospital, Tokyo, Japan, **2** Department of Clinical Medical Sciences, International University of Health and Welfare Graduate School of Medicine, Chiba, Japan, **3** International University Health and Welfare, Graduate School of Health and Welfare Sciences, Tochigi, Japan, **4** Department of Anesthesiology and Intensive Care Medicine, International University of Health and Welfare School of Medicine, Chiba, Japan

* 21M3008@g.iuhw.ac.jp

## Abstract

### Background

In Japan, the escalating demand for anesthesia services has resulted in a shortage of anesthesiologists, presenting a societal challenge. Urgent measures involve the imperative task shifting to nurses. The perspectives of anesthesiologists on perianesthesia nurses (PANs) and the PAN system significantly influence the collaboration between anesthesiologists and PANs.

### Methods

Twenty-four anesthesiologists initially approached in writing among a pool of 304, ultimately agreed to participate in interviews. Verbatim transcripts from these interviews were analyzed using the framework method. In this procedure, data analysis was facilitated by MAXQDA software (version 22.7.0) to construct a case-code matrix, enhancing the reliability of our findings.

### Results

Five themes and fifteen categories related to PAN and its system emerged. Participants provided insights into the diverse social conditions accompanying the implementation of the PAN system. They highly regarded PANs as colleagues, expecting a spectrum of capabilities. Nevertheless, the analysis revealed considerable variation in role expectations across institutions and individuals, with some perspectives being mutually critical. Conflicting opinions emerged on two crucial aspects: assigning anesthesia management conducted by PANs and substituting PANs for anesthesiologists. Multiple suggestions were put forth for enhancing and evolving the PAN system.

### Conclusion

This qualitative study is the first to reveal that Japanese anesthesiologists hold diverse perspectives on PANs and the system. The approach was well-suited for exploring diverse

**Data Availability Statement:** Data Availability: Providing access to the full interview records could potentially compromise the anonymity of the participants. While names of individuals and

institutions have been removed from these records, previous literature has pointed out that completely concealing identities from interview transcripts is virtually impossible [1]. The participants in this study were 24 out of 304 anesthesiologists who were full-time staff members at hospitals where PANs are employed. When short excerpts from the interviews used in the findings section of this paper are viewed in isolation, they do not reveal the identities of the participants. However, the full records contain more detailed information, including personal anecdotes and specific details about the participants' work environments, which could potentially allow readers familiar with the Japanese anesthesiology community to deduce the identities of the participants. Informed consent was obtained from each interviewee with an agreement that "when publishing the research results, we will not include any information that could identify the research subjects." The participants provided us with frank responses under the condition that we would maintain their anonymity. Providing complete interview records may violate our ethical responsibility to protect the participants' privacy by breaking anonymity. Researchers who wish to access the data underlying the research results should contact the Research Ethics Committee of International University of Health and Welfare (contact: rinri@iuhw.ac.jp) to inquire about the necessary procedures for access. The use of data must follow the instructions of the Research Ethics Committee and maximize the protection of participants' privacy and anonymity. [1] van den Hoonaard WC. Is Anonymity an Artifact in Ethnographic Research? Journal of Academic Ethics. 2003;1: 141–151.

**Funding:** The author(s) received no specific funding for this work.

**Competing interests:** The authors have declared that no competing interests exist.

perspectives, showing significant differences among institutions and individuals. Our data provided crucial insights, including findings suggesting potential barriers to task shifting of anesthesia duties to PANs.

## Introduction

In recent years, there has been a rapid escalation in the demand for anesthesiologists in Japan. With the impending transition to a super-aged society by 2025, where one in four citizens will be aged 75 or older, the number of patients requiring surgery continues to rise [1]. Furthermore, to oversee patients with intricate comorbidities, anesthesiologists are deemed essential in operating rooms and various medical specialties such as intensive care units, pain clinics, palliative care services, and emergency responses [2]. However, there is a need for more anesthesiologists to address the escalating demand, consequently leading to protracted waiting intervals for surgical procedures. In various regions, this shortage has compelled the delegation of anesthesia management to non-anesthesiologists [3]. Moreover, despite the growing demand for painless delivery, the scarcity of anesthesiologists to deliver this service persists, resulting in a painless delivery penetration rate in Japan that falls below 10% [4].

Delegating responsibilities to nursing professionals is a prevalent strategy to mitigate the shortage of anesthesiologists and the resultant decline in the quality of healthcare services [5]. Nurse anesthetists provide anesthesia care in numerous countries. Their responsibilities, tasks, and roles vary significantly worldwide due to diverse historical backgrounds and healthcare systems [6]. In Japan, there are no recognized positions equivalent to nurse anesthetists or anesthesia assistants. However, a graduate school of nursing initiated the training of advanced practice nurses to provide specialized assistance in anesthesia care in 2010 [7]. The objective was to educate nurses who, under the supervision of an anesthesiologist, would be engaged in perioperative care. The nurses were named "perianesthesia nurses" (PANs). In Japan, the term "perianesthesia" is used to emphasize the role of specialized nurses who support anesthesiologists in a wide range of anesthesia-related tasks. Several nursing graduate schools now provide PAN training programs, where students learn anesthesia-specific pathophysiology and clinical pharmacology. Upon graduation, they execute anesthesia-related responsibilities under the supervision of anesthesiologists within the framework of 'assisting medical treatment' defined by the Health and Medical Assistance Act [2, 8]. As of 2021, approximately 30 PANs are actively serving as anesthesia-related personnel in diverse locations, and both the number of PANs and the facilities they serve are steadily increasing each year [9].

A lack of understanding among doctors regarding the role and job description of nurses with the new specialty, or apprehensions about potential threats to their positions, may hinder establishing positive working relationships [10, 11]. Contrarily, when doctors trust in the competence of nurses and establish a robust working relationship, patient care duration can be reduced and healthcare costs can be minimized [12, 13]. For the successful implementation of task shifting, it is imperative that doctors feel secure, possess a clear awareness of the scope of practice associated with the new position, and foster an enhanced working relationship.

No studies have yet investigated the perspectives of anesthesiologists working with PANs regarding PANs and the PAN system. This study aims to present a detailed investigation of anesthesiologists' diverse perspectives on PANs and the PAN system. A deeper understanding of anesthesiologists' views may contribute to strengthening collaboration between anesthesiologists and PANs, potentially accelerating the development of the PAN system.

## Methods

### Qualitative approach and research paradigm

In this study, we adopted an exploratory qualitative approach to analyze in detail the perspectives of anesthesiologists, using data from participants' narratives. The qualitative research method focuses on individual experiences and emotions, providing a deeper insight into the current situation [14–16]. Our analysis was grounded in the "framework method." The framework method, created in the late 1980s for large-scale policy research, is a flexible tool adaptable to various qualitative approaches without a specific theoretical alignment [17]. It is one of the most suitable methodologies for analyzing interview data, eliciting themes through both intra-case and inter-case comparisons.

### Researcher characteristics and reflexivity

The first author (MT) has amassed 16 years of experience in anesthesiology and currently holds a supervisory role. Throughout the interviews, MT was mindful of variations in age and experience, directing efforts to establish an environment where senior residents and specialists could securely express their thoughts. The first author's extensive experience in anesthesiology allowed her to understand the participants' perspectives and ask appropriate follow-up questions. Accordingly, she could build rapport with participants and elicit rich responses. She lacked decision-making authority concerning the nature of PAN's work or its institutions, enabling her to engage in honest and open discussions with all participants. The second author (SK), a prominent qualitative researcher, oversaw all data analysis. The third and fourth authors (YB and KK), supervisors with approximately 30 years of anesthesia experience, possessed extensive knowledge of the PAN system.

### Context

At the time of initiating this study in December 2021, there were five graduate school programs in Japan training PANs, located in various regions, including the Tokyo metropolitan area, Kansai, and Chubu. Graduates were working in anesthesia-related roles at their alma mater university hospitals, other university hospitals, and general hospitals alongside anesthesiologists with diverse experience levels and backgrounds. One graduate school that had initiated a PAN training program had already discontinued it. However, its affiliated hospital continued to employ PANs and assign them anesthesia care responsibilities. PANs' duties vary based on the requirements of each facility, including pre-operative outpatient care, intra-operative anesthesia management, post-operative pain management, non-operating room anesthesia, and pain-free delivery assistance [18, 19]. PANs' work is carried out in accordance with the law regulating nursing work in Japan (the Act on Public Health Nurses, Midwives, and Nurses) within the framework of "medical treatment assistance" as defined by this legislation. However, there are no unified standards or laws set by the government or relevant associations and academic societies regarding certification, educational standards, or the responsibilities and authority of PANs. Their scope of practice is left to the discretion of each employing institution [9].

### Participants and sampling

We identified 16 hospitals employing PANs through email inquiries to five graduate schools and by reviewing programs of major anesthesia-related conferences held from 2018 to 2020. These include 5 main hospitals, 3 affiliated hospitals, 3 university hospitals without graduate schools offering perianesthesia nursing programs, and 5 general hospitals. Between December

2021 and January 2022, we emailed the heads of anesthesiology departments at these 16 hospitals, explaining the study's purpose and privacy protection measures. We requested permission to survey all full-time anesthesiologists in their departments to gauge interest in study participation. Part-time anesthesiologists were excluded due to limited interaction with PANs. The survey was conducted using paper forms. Participants willing to cooperate in face-to-face or online interviews about PANs were asked to provide their names, positions, and email addresses, and return the form to the researchers by mail. We confirmed with each department head that 304 surveys were distributed.

Thirty-seven anesthesiologists expressed their desire to participate in the study. Following initial correspondence clarifying the study's content, multiple anesthesiologists withdrew; two cited concerns about the relevance of their experiences to the study's focus, three scheduling constraints, and eight refused our contact. Ultimately, twenty-four anesthesiologists participated in the interview survey. Despite the substantial time commitment required for interviews, these participants volunteered to share their experiences and perspectives, confirming they had rich insights and a solid motivation to contribute to the study. This self-selection process was advantageous, yielding participants who invested their time and provided detailed and diverse perspectives on working with PANs. The participants were affiliated with nine distinct hospitals. Five of these hospitals had graduate courses in perianesthesia nursing. Interviews were conducted from January 2022 to May 2023. Participant characteristics (tenure, position, and gender distribution) are presented in Table 1.

## Ethical issues pertaining to human subjects

The study was conducted with the approval of the Ethics Committee of the International University of Health and Welfare (Approval No.21-Im-056, Registration date: September 30, 2021).

The study's purpose, objectives, and content were explained to all participants via written materials. For face-to-face interviews, informed consent was obtained in person prior to the interview. For online interviews, consent forms were sent and returned via mail before proceeding with the interview. In both cases, participants were informed of the voluntary nature of their participation and their right to withdraw anytime without consequences.

Due to the COVID-19 pandemic, interviews were conducted either face-to-face with infection control measures (e.g., mask-wearing, ventilation) or online via video conferencing, based on participants' preferences. All interviews, regardless of format, were held in quiet, private spaces. For video conferences, both parties confirmed their private settings, ensuring the protection of confidential information, mirroring the privacy standards of face-to-face interviews. All interviews were recorded (audio or video) with prior consent. The recorded data was securely stored and strictly managed. All interviews were transcribed verbatim in Japanese, with identifiable information excluded from the transcripts to ensure participant confidentiality. Recordings will be permanently deleted upon study completion.

**Table 1. Characteristics of study participants (N = 24).**

| Characteristic | Details |
| --- | --- |
| Total participants | 24 |
| Tenure in current position | Median: 20.5 years (range: 2–30 years) |
| Position (number of participants) | Senior residents (3), Specialists (4), Supervisors (17) |
| Gender | Male: 18 (75%), Female: 6 (25%) |

This table summarizes key characteristics of the study participants.

## Data collection methods, instruments, and technologies

Data were gathered through unstructured interviews, conducted in person or via video conferencing from January 2022 to May 2023. 17 of 24 participants preferred online interviews. These were performed using a secure video conferencing platform (Zoom, Zoom Video Communications, Inc., San Jose, CA, USA). All interviews were audio-recorded with consent. A digital recorder was used for face-to-face interviews, while online interviews utilized the platform's built-in recording function. MT conducted all interviews in Japanese.

The initial prompt was, "Could you please outline the primary responsibilities of perianesthesia nurses in your hospital?" This approach aligned with our objective of acquiring a comprehensive narrative. As participants recalled their daily emotions toward PAN, the interviewer listened to their narratives and encouraged them to expand on their stories. To deepen the discussion, a set of open-ended questions was used as deemed appropriate (S1 Appendix). Participants responded to various topics related to PANs, including their diverse roles, responsibilities, PAN system's current state, and its challenges. The management of anesthesia facilitated by PANs, hereafter denoted as PAN anesthesia, emerged as a topic of significant interest during the interviews. The median interview duration was 55 minutes (37–96 minutes). Each participant was interviewed once, with no follow-up interviews conducted. Participants were allowed to review their interview transcripts, but none chose to exercise this option.

## Data analysis

We analyzed the interview data using the framework method. This process consisted of six stages: familiarization with the interview, coding, initial analytical framework creation, iterative analysis, final analytical framework creation, analysis using a software (Fig 1).

Familiarization involved reviewing transcripts and re-listening to audio recordings. Coding incorporated words to characterize each passage precisely [20, 21]. The initial analytical framework was established after coding for the first eight participants and iteratively adjusted as new codes emerged in subsequent interviews. The constant comparative method effectively generated categories [22].

Saturation of the theme was achieved by the 23rd participant, with the 24th interview exploring potential unidentified topics. The final analytical framework comprised five distinct themes, each accompanied by a note containing categories, descriptions of codes, and keywords signifying relationships between themes. This framework was endorsed after consensus among researchers (S2 Appendix).

The analysis using MAXQDA Plus 2022 (release 22.7.0, build 230619), software designed for qualitative data analysis, efficiently showed which part of the analysis framework corresponded to each sentence of the 24 transcripts (S3 Appendix). By performing this process on all data, we ensured comprehensive coverage and prevented data omission.

Finally, we created a "case-code matrix" with rows representing cases (24 participants), columns representing codes (themes I-V and corresponding categories) (Fig 2). Each matrix cell is directly linked to its respective data, and we can reference the data itself as required (Table 2). This analytical process effectively condensed the entire dataset without sacrificing individual context. This facilitated a more profound analysis, as the data can be readily compared within individual cases and between cases while preserving the contextual integrity of the data. The compressed dataset also enhances the reliability of data analysis through collaborative analysis among multiple researchers.

This table presents a focused excerpt from the larger case-code matrix created with MAXQDA, showing the connection between individual cases and the codes derived from the data. Each cell offers a direct link to the original interview excerpts, highlighting the alignment between the qualitative analysis and the specific narratives of each case. The arrangement of

**1. Familiarization with the interview**
- Iteratively reviewing the transcripts
- Re-listening to selected or all audio recordings
- Annotating our impressions in the margins of the transcripts

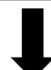

**2. Coding** (the fundamental analysis in qualitative research)
- Incorporating words (codes) to characterize each passage precisely
- Up to 8th participant

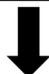

**3. Initial analytical framework creation**
- Grouping codes with common elements
- Assigning the groups to "concepts" that precisely describe their shared characteristics
- Organizing multiple "concepts" into "categories"
- Illustrating the relationships among them

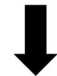

**4. Iterative analysis: 9th to 24th participant** (saturation at 23rd)
- Coding → Identifying new codes → Framework adjustment utilizing the continuous comparison method

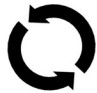

**5. Final analytical framework creation**

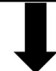

**6. Analysis using a software (MAXQDA Plus 2022 )**
- Importing final analytical framework and 24 transcripts
- Viewing textual data and list of codes simultaneously
- Assigning appropriate codes to each sentence for all 24 participants' data
- Creating and analyzing Case-code matrix

**Fig 1. Data analysis process.** This figure illustrates the six-stage qualitative data analysis process using the framework method. It depicts the progression from familiarization with interview data through coding, framework creation, and iterative analysis, culminating in software-aided final analysis.

cases and codes has been formatted to enhance the readability of the table. The notion "(q75)" follows an excerpt, denoting that it is the 75th quote in the interview data.

The inductive process was executed by a single researcher, MT, under the supervision of a qualitative research specialist, SK. Researchers YK and KK mitigated bias by offering coding guidance and scrutinizing the framework and figures creation process.

The report follows the Standards for Reporting Qualitative Research (SRQR). A completed SRQR checklist is attached in the S4 Appendix [23].

## Results

Five common themes emerged associated with anesthesiologists' perspectives on PAN: I) Anesthesiologists' perspectives on the implementation of the PAN system in Japan, II) High

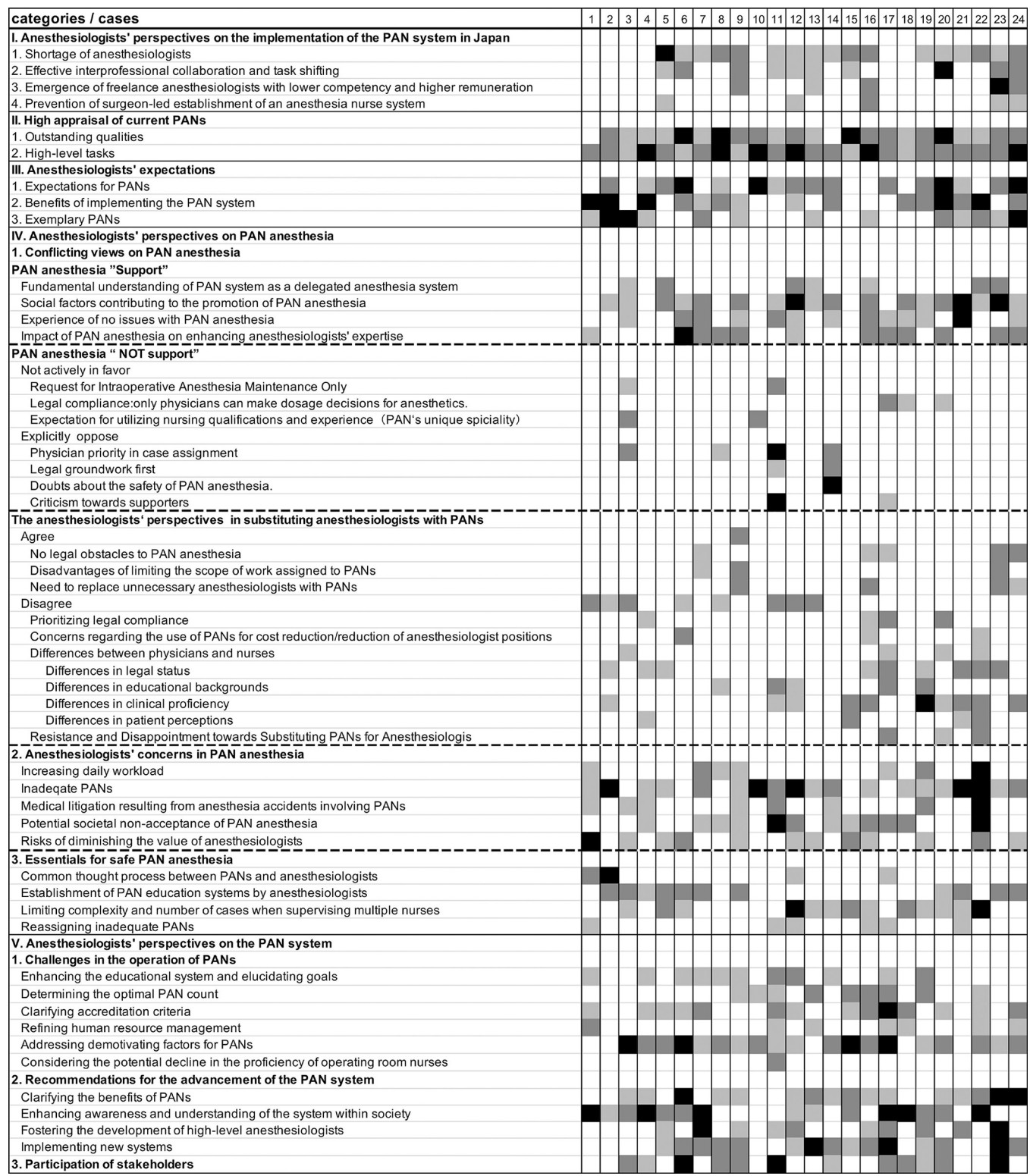

**Fig 2. The case-code matrix.** This figure presents the case-code matrix created in MAXQDA and visualized in Excel. The shading of each cell indicates the frequency of mentions for each theme. White signifies a count of 0, light gray denotes a count of 1, dark gray indicates a count ranging from 2 to 4, and black represents a count of 5 or more. This color-coding system facilitates a quick, at-a-glance assessment of the occurrences of codes across the 24 cases analyzed.

**Table 2. Data linkage through the case-code matrix.**

| | IV.1.Conflicting views on PAN anesthesia / PAN anesthesia "Support" |
|---|---|
| | **Social factors contributing to the promotion of PAN anesthesia** |
| **case14** | 1) A: Anesthesia administered by a nurse is not preferable, but I personally don't want to emphasize it too strongly. |
| | Q: Why don't you want to emphasize it? What is the reason for not wanting to say it strongly? |
| | A: Because there's no other choice. (q75-77) |
| | 2) A: In places like Tokai or Tohoku, despite the anesthesia physician numbers not being significantly different, I imagine the case load would be high. There might be situations like that. But in front of those doctors, you can't just say, "No way," it's really hard to say. |
| | Q: Reality. |
| | A: Yeah, realistically, it is what it is. But when you think about how to ensure safety, people who have received reasonable training are much better than trainee doctors who don't know anything. We've seen that they can intubate confidently, even better than other trainee doctors. In that sense, it might be inevitable. (q79-81) |
| | 3) I feel like having a perioperative nurse do it is ten times better than having a doctor who hasn't done or doesn't know anything. (q180) |
| **case16** | 1) Well, you know, when it comes to administering anesthesia, it's so much better when someone who has been properly trained does it, compared to doing it solo. (q32) |
| | 2) Without them, we just can't carry out certain tasks, you know? (q111) |
| **case22** | 1) Yeah, that's true. Exactly. So, it really depends on the number of anesthesiologists. I guess from the hospital's perspective, they might think that salary-wise, perianesthesia nurses could help keep costs down. (q195) |
| **case23** | 1) In this particular hospital, actually, medical office assistants handle a significant portion of the interviews and tasks. They cover a lot, even detailed information. They grasp everything, including medical history and all the medications they're taking. Some people expect perioperative nurses to handle that level of responsibility, but in reality, it's done by the office staff. (q47) |
| | 2) Q: Do they not want to be used at the discretion of the anesthesiologists? |
| | A: No. |
| | Q: So, how would they prefer to be utilized? |
| | A: First of all, they find it fulfilling to be able to handle a full anesthesia case properly, you know? They don't see themselves as just nurses; they consider themselves "little doctors". I treat them that way too, and they, in fact, take pride in being little doctors. (q52-55) |
| | 3) The number of surgeries is constantly increasing. They say, "We need more anesthesiologists!" The number of female doctors is growing. "We need more anesthesiologists!" They want to take on new tasks in pain management. "We need more anesthesiologists!" Everyone is asking for more, for various reasons. But we can't provide what we don't have. If we continue like this, it'll probably last forever, at least for several decades. Hardly any country in the world restricts nurses from doing anesthesia–except Japan. So, why not just go ahead and introduce it here, you know? (q227) |
| | 4) I mean, I'm not exactly sure what the playbook says on this one, but it feels like the best move is to just let it spread far and wide, let folks have a go at it freely, and then pull back on anything that doesn't pan out, right? (q167) |
| | 5) Q: Did the perioperative colleagues you work with consciously share the frustration towards freelance anesthesiologists and say, 'Let's kick them out together'? |
| | A: Well, yeah, that's there. I think everyone, or at least a good number of them, feels that way. I think they share my sentiment. (q275-276) |

appraisal of current PANs, III) Anesthesiologists' expectations, IV) Anesthesiologists' perspectives on PAN anesthesia, and V) Anesthesiologists' perspectives on the PAN system (Fig 3). The figure illustrated nine arrows: eight indicated "influences" from one theme to another, and one denoted a "similarity" between two themes. In Theme IV, conflicting viewpoints were observed. The details of their nature are described in the following sections. We have included detailed tables in the S5 Appendix to provide a comprehensive overview of our analysis. These tables present the 5 main themes, their associated categories, and subcategories, along with

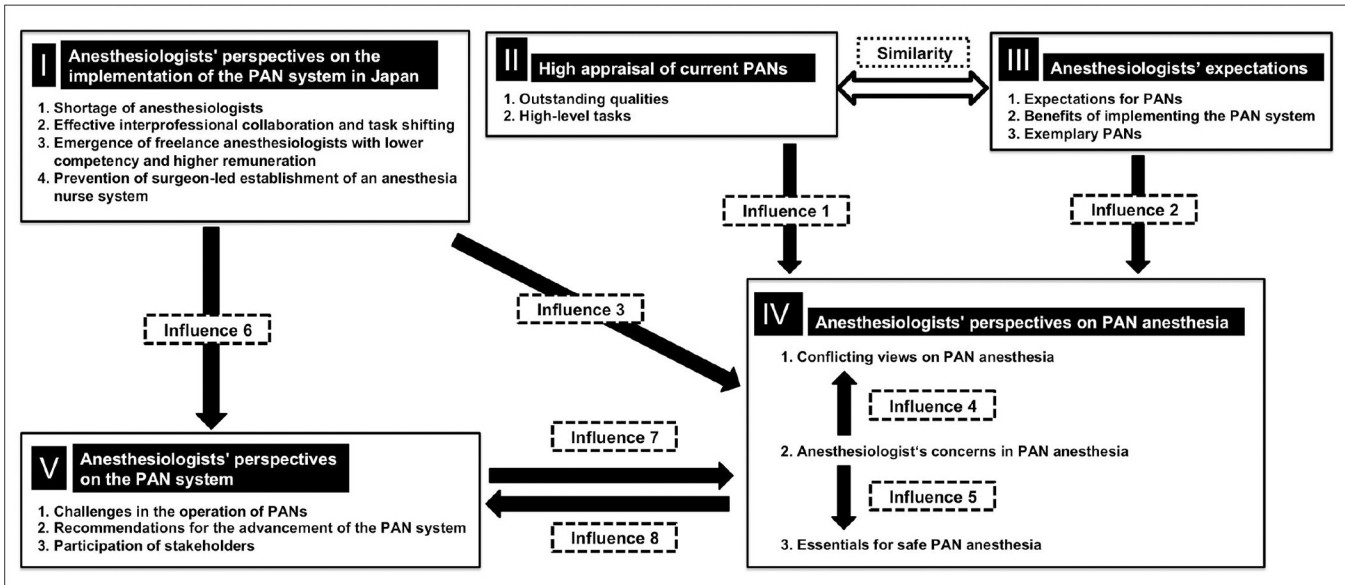

**Fig 3. Structure of interrelated themes revealed in anesthesiologists' perspectives on PANs.** This figure shows how the five themes are interconnected. Arrows indicate the influence from one theme to another, and a two-way arrow highlights a similarity between two themes.

representative quotes and reference numbers for each code. One of the tables from S5 Appendix is used later in this section, as Table 3, to illustrate our analytical process in depth.

## I. Anesthesiologists' perspectives on the implementation of the PAN system in Japan

Participants highlighted four social backgrounds for introducing the PAN system: 1. Shortage of anesthesiologists, 2. Effective interprofessional collaboration and task shifting, 3. Emergence of freelance anesthesiologists with lower competency and higher remuneration, and 4. Prevention of surgeon-led establishment of an anesthesia nurse system. Anesthesiologists have observed a shortage, attributing it to their hectic daily schedules and the growing complexity of their work. Additionally, they expressed expectations for task shifting to nurses, which were influenced by the recent rise in the importance of care during the perioperative period. They highlighted that freelance anesthesiologists with lower competency and higher remuneration have emerged as a social issue in Japan. Furthermore, they mentioned concern about the possibility of surgeons independently training their nurses for anesthesia management, emphasizing the need for anesthesiologists to prepare PANs before such a transition. To accurately reference interview excerpts, citations are formatted by case and quotation number, such as (9/q87) for the 87th quote from Case 9. To enhance readability and conciseness of the research, interview quotations have been carefully condensed to eliminate conversational redundancies without altering the original meaning.

> "In community hospitals and others, they're severely understaffed, with COVID-19, many anesthesiologists were taken up by the ICU. they had no choice but to go there. Who's left to handle anesthesia in the operating rooms?" (9/q87)

> "The Japan Society of Anesthesiologists or similar organization should handle perianesthesia nurse registration. They must be placed with anesthesiologists as working alone with surgeons would be a huge problem." (16/q163)

**Table 3. Anesthesiologists' concerns about PAN anesthesia.**

| subcategory | Interview/ quote no. | code | quote summary |
|---|---|---|---|
| Increasing daily workload | 6/q54 | Assigning cases requires careful consideration | Sedation outside the operating room is typically administered by a perianesthesia nurse working alongside an anesthesiology instructor. Despite the need for advanced skills, there are limited opportunities for senior residents to participate and gain experience. I find this concerning. |
| | 9/q61 | Educational burden | There are numerous individuals requiring education, including not only PANs but also medical students, interns, and residents. There aren't sufficient anesthesiologists available to instruct them. |
| | 22/q292 | Burden of simultaneous supervision | To efficiently utilize multiple PANs in the operating room, simultaneous anesthesia is necessary. While there are managerial benefits, the burden on the supervising anesthesiologist is increasing. |
| Inadequate PANs | 22/q170 | Underestimating anesthesia / Lack of apprehension | I am concerned about PANs just focusing on the monitor, not recognizing that "something minor can be fatal" and assuming "nothing will happen anyway". |
| | 10/q108 | Depend on manuals | It is unfortunate PANs do not take advantage of their own abilities, saying things like, "That's not my job," or "The doctor told me to do it, so I'll make a manual to handle it," or "If that's what the doctor tells me to do, I won't do that job anymore. |
| | 12/q317 | No responsibility as a nurse | Now is the time for nurses to assume responsibility. It is regrettable that they are irresponsible as "the physician is responsible for medical care." They just wait for the doctor's orders, do nothing even if the patient is suffering unless the doctor tells them to, etc. |
| | 21/q318 | Mistaking oneself as a doctor | There may be Perianesthesia Nurses who mistakenly see themselves as doctors, thinking they are great and amazing because they can administer anesthesia. |
| | 18/q145 | No guarantee that the best will always enter | The PANs who are currently working with us are serious people who have carved out their own paths. Ordinary people would be hard-pressed to have that level of motivation. |
| | 24/q81 | Difficult to make all PANs ideal through education alone | It is impossible to elevate all PANs to a high level through education alone. Personal aptitude, character, and motivation also have a significant impact. |
| Medical litigation resulting from anesthesia accidents involving PANs | 9/q105 | No legal backing | It is a fact that Perianesthesia Nurses are administering anesthesia without legal backing. |
| | 4/q133 | No legal knowledge | I am not familiar with the laws related to liability in case of problems and other associated aspects. |
| | 19/q86 | No legal precedent | You do anesthesia with a PAN, an apparent anesthesia accident comes along, and then the case goes to court. Unless we see the case law, we don't know if there is any difference compared to similar accidents done with residents. I think this will happen in the future. Anesthesiologists are anxious. All anesthesiologists are afraid of the trial. |
| | 19/q76 | Responsibility lies with the anesthesiologist | Let's assume that the anesthesia incident happened because of the inadequacy of the Perianesthesia Nurse. I believe the responsibility should be shouldered by the anesthesiologist who was supervising the PAN. |
| | 9/q114 | Being a nurse is a disadvantage in lawsuits | In a medical lawsuit, the question may be asked, 'Why were there nurses in the operating room instead of doctors in the first place?' In that situation, the fact that the nurse was a very studious nurse might not be taken into consideration. The world has not yet fully accepted nurses who can administer anesthesia. |
| Potential societal non-acceptance of PAN anesthesia | 22/q116 | Regional differences in acceptance | In a rural hospital where there are no anesthesiologists, the acceptance of nurse-administered anesthesia might be more plausible. Even though they are not anesthesiologists, they are individuals who have received proper anesthesia training. However, in urban areas where there is no shortage of anesthesiologists, the absence of an anesthesiologist in the operating room can become a factor contributing to issues or provide a reason for patients to blame us. |
| | 12/q111 | Negative feelings of patients | When something goes wrong, the fact that it was done by a nurse rather than a doctor could evoke negative feelings in the patient. |

(*Continued*)

**Table 3.** (Continued)

| subcategory | Interview/ quote no. | code | quote summary |
|---|---|---|---|
| Risks of diminishing the value of anesthesiologists | 15/q89 | Risk of making anesthesiologists unnecessary | One of the causes of anxiety is the notion that 'the nurses are excellent and can manage anesthesia well, so there is no need for an anesthesiologist anymore. |
| | 5/q66 | Risk of damage to value of anesthesiologist | There is a potential for interns to perceive that 'administering anesthesia is a task that nurses can handle, and it's not a task for us physicians. |

## II. High appraisal of current PANs

This theme encompasses two categories: 1. Outstanding qualities and 2. High-level tasks. Anesthesiologists highly praised the character and capabilities of their PAN colleagues, recognizing outstanding qualities such as diligent work, heightened motivation, superior communication skills arising from mature personalities, academic thought processes, and unique nursing-oriented cognition. Furthermore, PANs were evaluated for their proficiency in precise intraoperative anesthesia management and preoperative and postoperative tasks, showcasing their high-level functions within the anesthesia care team. This positive appraisal extended to their comprehensive understanding of anesthesia and their elevated involvement in various anesthesia services.

"The PANs think independently, and don't overly rely on us. They confirm necessary things and keep us informed through reporting, communication, and consultation. I've never experienced feeling scared that they executed something without consulting us." (3/q19)

"The perianesthesia nurse can administer anesthesia, call for help if issues arise, and participate in education. Our perianesthesia nurse is ideal." (7/q152)

## III. Anesthesiologists' expectations

Anesthesiologists articulated varied expectations regarding PANs and the PAN system. This theme encompasses three categories related to anesthesiologists' expectations: 1. Expectations for PANs, 2. Benefits of implementing the PAN system, 3. Exemplary PANs. They expressed the expectation that the involvement of PANs in diversified anesthesia services would reduce the workload of anesthesiologists. This involvement was expected to enable anesthesiologists to engage in more advanced work and improve the quality, safety, and patient satisfaction of anesthesia care. The "anesthesia-specialized nurses" in Japan were appraised as "unique." Their unique responsibilities were identified as follows: collaborating with anesthesiologists, enhancing cooperation between anesthesiologists and other medical practitioners and departments, assuming leadership and supervisory roles over other nurses, expressing perspectives from both anesthesiologists and nurses, ensuring seamless patient care from the preoperative to postoperative period, and applying their care skills. Anesthesiologists identified exemplary PANs as possessing a high degree of interpersonal skills and providing excellent anesthesia assistance, similar to the elements outlined in Theme II (Fig 3). Additionally, four ideal mindsets were traced: an attitude of enjoying the business, a solid eagerness to learn, understanding of the nurse's position ("*wakimae*"), and non-monetary motivations.

"Nurses' work is probably really important, and we have various things to think about in the operating room. Without time for mutual conferences, we need to understand more about what each other is thinking while working. . . Perianesthesia nurses likely best

understand what doctors and nurses are lacking. They're a unique profession. I think it would be great to have quite a few people excelling in that role." (10/q66)

"Once properly trained, PANs can handle many preoperative evaluations. Our work involves quite a lot of these things, actually. So, to sum it up, we want to gradually shift anesthesiologists' resources towards more critically ill patients and core anesthesia-related tasks."(6/q98)

"It's *Wakimae*. . . that's what it's about, they're not anesthesiologists after all." (21/q326)

## IV. Anesthesiologists' perspectives on PAN anesthesia

This theme encompasses three categories: 1. Conflicting views on PAN Anesthesia, 2. Anesthesiologists' concerns in PAN anesthesia, 3. Essentials for safe PAN anesthesia. The details of their nature are described in the following sections.

### 1. Conflicting views on PAN anesthesia

Conflicting opinions arose regarding the endorsement of PAN anesthesia (Fig 4). Anesthesiologists endorsing PAN anesthesia contended that the PAN system is synonymous with PAN anesthesia. They substantiated their argument by referencing various social factors: the shortage of anesthesiologists, simplified anesthesia management facilitated by pharmaceutical advancements, anesthesia nurses in other countries, and public demand for more anesthesia cases with reduced wages and a smaller workforce. The favorable assessment of the current PAN in anesthesia management, as indicated in Theme II in Fig 3, strengthened the endorsement of PAN anesthesia (Influence 1). Moreover, they claimed that PAN anesthesia would enhance anesthesiologist competence and medical safety.

"Anesthesiologists can shift their focus to patients with severe conditions. And more, they can dive into intensive care, join rapid response team, manage labor pain, and dealing with sedation stuff outside the operating room. It's a way to broaden the anesthesiologist role.

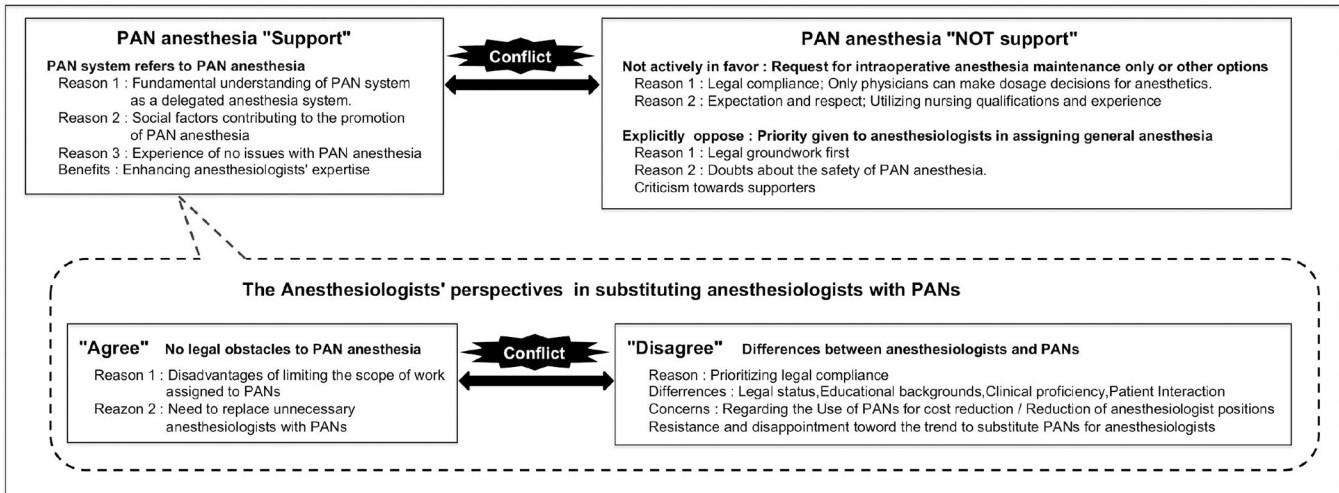

**Fig 4. Two conflicting perspectives on PAN anesthesia.** This figure shows dual conflicts within anesthesiologists' perspectives: one between proponents and opponents of PAN anesthesia and another concerning substituting anesthesiologists with PANs. Each stance is accompanied by texts detailing their respective rationales.

It's not just about the surgery room. We're taking care of things all over the hospital, being on point with the Rapid Response Team and contributing to improving overall hospital safety." (6/q97)

However, it was noted that not all anesthesiologists perceive the "enhancement of anesthesiologists' competence" as a benefit.

"Nurses entering the field will be increased competition. I believe this may require anesthesiologists to exert more effort. I can comprehend their emotional resistance, fearing when life events like childbirth or childcare align, they might find no place to return to in the workplace." (17/q224)

They asserted that non-PAN ordinary nurses could carry out the preoperative and postoperative tasks requested by anesthesiologists opposing PAN anesthesia. Furthermore, they expressed the opinion to the anesthesiologists in opposition that assigning low-complexity anesthesia duties to PANs was a default and a matter that should be acknowledged. Anesthesiologists who do not support PAN anesthesia also conveyed some acknowledgment of the social factors contributing to the promotion of PAN anesthesia.

Anesthesiologists who "do not support" PAN anesthesia were divided into two groups: those who are "Not actively in favor" and those who "Explicitly oppose." The former group tolerated intraoperative anesthesia maintenance duties by PANs but resisted nurses using their judgment in administering drugs, citing legal grounds. Others requested PANs to engage in different duties rather than anesthesia management. This was due to respect and expectation for the unique skills only PANs can perform, as indicated in Theme III (Influence 2).

"I want PANs to have at least minimum anesthesia management skills, but prefer they focus on perioperative support. I wish PANs could offer a bit more assistance with preoperative patient orientation and tasks that anesthesiologists can only do minimally." (3/q82)

"I feel like there's stuff that only a PAN can handle, things that fall right in between a nurse and an anesthesiologist." (10/q50)

The latter considered anesthesia practice a medical specialty only physicians are allowed to practice. They also expressed concerns about the safety of PAN anesthesia and criticized the current situation where PAN anesthesia operates, relying only on institutional certification without any legal framework.

"I feel like having a nurse handle anesthesia is a bit of a legal gray area. And, safety-wise, too. . .(omission) I reckon dedicated anesthesiologists should give anesthesia." (14/q73, q176)

"I can't trust 'self-proclaimed' or 'institutional certification holders' without government endorsement. It means they lack the qualifications, and I feel sorry for PAN if something goes wrong. I can't take responsibility. PANs are like children who are great at driving but don't have licenses. In a real emergency where I'm unconscious or can't move, I might ask them to drive, but I can't let them routinely administer anesthesia." (11/q128)

Even among anesthesiologists supportive of PAN anesthesia, conflicting opinions emerged regarding substituting anesthesiologists with PANs. Anesthesiologists supporting the proposal

asserted that the current law could be interpreted to permit nurses to practice medicine as long as a physician directly supervises them. They contended that PANs could perform nearly as effectively as senior residents. Recognizing that restricting PANs to minor cases would not address the shortage of anesthesiologists, they urged for a proactive delegation of anesthesia duties, advocating for removing restrictions on handling severe cases based on their capabilities. Additionally, they regarded as "unnecessary" those anesthesiologists who lacked the profession's unique skills and ambition and who were content with doing work that nurses could do, strongly advocating for their replacement by PANs. Underlying this perspective was a significant irritation regarding the "emergence of freelance anesthesiologists with lower competencies and higher remuneration," as delineated in Theme I (Influence 3).

> "I want to replace more freelance anesthesiologists with nurses. I want people who don't work to leave (omission 1). They're just after money. I'm not saying they shouldn't make money, but if they want money, they should also take responsibility (omission 2). I aim to gradually phase out positions for freelance anesthesiologists with low aspirations, who only want to do their own anesthesia, not mentoring younger staff or being involved in various organizational operations." (9/q75, q216, q217).

In contrast, anesthesiologists opposing the proposal cited four reasons why PANs and physicians cannot be equal: differences in legal status, differences in educational backgrounds, differences in clinical proficiency, and differences in patient perceptions.

> "In medicine, everything revolves around the law. Personally, having a nurse perform a medical procedure. . . This stance won't shift in the future, all for patient safety. Some things remain a no-go, regardless of evolution." (17/q208)

> "Knowledge levels vary for sure. Some PANs study hard, but the foundations of what they've learned are distinct." (12/q248)

> "Even if PANs possess the knowledge, decision-making ability is a separate matter. I think only doctors can really handle the decision-making part." (21/q252)

> "Even if it's not directly caused by someone's actions, there's a chance something might happen during the perioperative period. In JAPAN today, this may lead to scrutiny of work nature. If a doctor acted, it's acceptable. However, if a nurse, even if qualified, did it, and something happened. . . Even if consequential, the patients might become aggressive." (4/q133)

Otherwise, they mentioned worries about reducing anesthesiologist positions due to the ongoing replacement of anesthesiologists and the potentially detrimental effect on the "uniqueness" of PANs, as shown in Theme III.

> "The PAN will become just another 'anesthesia machine.' It's disappointing for them. They're here to handle more perioperative care and nursing, not just to increase the case numbers." (6/q156)

## 2. Anesthesiologists' concerns in PAN anesthesia

Not only anesthesiologists who opposed PAN anesthesia but also those who expressed support for it raised several concerns related to PAN anesthesia. These concerns encompass

increased daily workload, inadequate PANs, medical litigation resulting from anesthesia accidents involving PANs, potential societal non-acceptance of PAN anesthesia, and risks of diminishing the value of anesthesiologists (Table 3). Vigilance against inadequate PANs and medical litigation reinforced the opposition to substituting anesthesiologists with PANs. Furthermore, they argued that the potential widespread societal perception that PANs could replace anesthesiologists could diminish the value of anesthesiology. This caution has led to opposition to anesthesiologists who advocate replacing anesthesiologists with PANs (Influence 4).

This Table provides a detailed view of anesthesiologists' concerns regarding PAN anesthesia, including relevant quotes and codes from the interview data.

### 3. Essentials for safe PAN anesthesia

Anesthesiologists identified four elements necessary for safe PAN anesthesia: common thought process between PANs and anesthesiologists, establishment of PAN education systems by anesthesiologists, limiting complexity and number of cases when supervising multiple nurses and reassigning inadequate PANs. The alertness to "the presence of inadequate PANs" indicated in subcategory 2 was linked to their "reassigning inadequate PANs" (Influence 5).

"Being able to work while we sort of understand what each other is thinking, the ideal situation." (2/q42)

"Some aspects of anesthesia remain tough to handle, maybe a few percent, even after years. If PANs don't recognize this, they might think, 'We can handle anesthesia alone.' They need to experience the anxiety that comes with anesthesia. We also need to let them know that." (5/q131)

"You'd expect the ability and areas where someone can take responsibility to kind of define themselves naturally. Normally, we monitor two people. In a pinch, we might handle three." (18/q92)

"I've got my boundaries, and I want PAN to stick to them. If PAN crosses them easily, I think, 'Uh-oh, I'm in a pickle.' If it's hard to manage, then I can't leave it to that PAN." (21/q163)

## V. Anesthesiologists' perspectives on the PAN system

Anesthesiologists have identified six challenges in the operation of PANs: enhancing the educational system and elucidating goals, determining the optimal PAN count, clarifying accreditation criteria, refining human resource management, addressing demotivating factors for PANs (such as the need for improved compensation), and considering the potential decline in the proficiency of operating room nurses due to the prioritization of PANs. In particular, they argued that addressing the demotivation of PANs is imperative, as it directly correlates with PAN turnover.

The anesthesiologists proposed four recommendations for advancing the PAN system: clarifying the benefits of PANs, enhancing awareness and understanding of the system within society, fostering the development of high-level anesthesiologists, and implementing new systems. One of the requested new systems was "a public system requiring the involvement of an anesthesiologist for PAN anesthesia." This request was influenced by the "Prevention of surgeon-led establishment of an anesthesia nurse system" indicated in Theme I (Influence 6).

Enhancing awareness and understanding of the system within society was closely linked to Theme IV's second category: the anesthesiologists' concerns in PAN anesthesia.

One of the survey findings indicated that certain anesthesiologists working with PANs lack a comprehensive understanding of the nature of PANs and the PAN system. The cause of this lack of knowledge stemmed from the absence of uniformity in defining the roles and responsibilities of PANs nationwide, along with ambiguity regarding the "scope of work that nurses can perform under the direction of a physician" and "which tasks assigned to PANs might violate of the law." This deficiency in comprehension heightened concerns about PAN anesthesia (Influence 7). They proposed that not only the societal aspect but also a multitude of anesthesiologists not involved in the program's initiation should be provided with the opportunity to improve their understanding of the PAN system.

> "I want a well-organized system. (omission) Maybe it's just me, but I might not fully understand the PAN system. It would be easier if they clearly set boundaries on how much work to trust PAN with?' The current system is unclear, and I'm uncertain about its limits. I don't know. . . " (1/q266,q271)

Moreover, the patients' emotional hesitancy to embrace PANs, as delineated in Theme IV, emerged as a risk capable of diminishing PAN motivation. This formed the rationale for advocating enhancing the public awareness and comprehension of the system (Influence 8). They articulated the challenges in addressing these issues solely through the endeavors of anesthesiologists and PANs, urging the participation of stakeholders such as the Japanese Society of Anesthesiologists, the Nursing Association, and governmental bodies.

## Discussion

This study revealed that anesthesiologists in Japan hold diverse perspectives on PANs. Anesthesiologists believed there were multiple compelling reasons for implementing the PAN system to be considered a social necessity. They highly valued PANs they directly worked with, recognizing them as capable partners expected to fulfill various roles in the future. However, perspectives on PANs' roles and scope of practice varied significantly among institutions and individuals, with some critical attitudes observed. Furthermore, anesthesiologists expressed various concerns about PAN anesthesia and the PAN system. They pointed out the difficulty of maintaining and developing the PAN system solely through the efforts of PANs and anesthesiologists, suggesting essential elements necessary for ensuring safety and advancing the system.

The anesthesiologist perceived four underlying factors for implementing the PAN system. "Shortage of anesthesiologists" and "effective interprofessional collaboration and task shifting" are widely recognized determinants for implementing the PAN system. However, "emergence of freelance anesthesiologists with lower qualifications and higher remuneration" and "prevention of surgeon-led establishment of an anesthesia nurse system" were newly identified in this study. In 2017, the Japan Surgical Society provided a report on the management of general anesthesia by surgeons in numerous Japanese hospitals. Simultaneously, the Society expressed dissatisfaction with the substantial fees earned by freelance anesthesiologists and the laxity observed in their anesthesia management. Additionally, the report highlighted instances where nurses were found to be administering anesthesia under the guidance of surgeons [24, 25]. The Japanese Ministry of Health, Labor, and Welfare has also expressed strong dissatisfaction with freelance anesthesiologists [26], and the comments provided by the anesthesiologists in this study are likely influenced by this situation.

The PANs received predominantly positive evaluations and were deemed deserving of the designation "exemplary." One anesthesiologist employed the uniquely Japanese term "*wakimae*" in discussing the mindset of PAN to convey his hopes and desires. "*Wakimae*" signifies being mindful of one's attitude and behavior in accordance with the other person or situation, as well as understanding one's own position and role and acting accordingly [27]. "Knowing your limit" is already acknowledged as a crucial skill for nurse practitioners [28]. However, the Japanese concept of "*wakimae*" adds a nuanced aspect of "expressing in a nonverbal and unobtrusive manner" that one possesses that skill. Japanese anesthesiologists sought PANs with a high level of competence in performing their duties but also an enjoyment of their work, a commitment to continuous learning, and an attitude of not prioritizing compensation. They appreciated the current PANs who have attained these qualities. Conversely, PANs lacking "*wakimae*" were considered inadequate.

The gender ratio of PANs might contribute to their perceived "*wakimae*". While less than 10% of nurses in Japan are male, approximately 30% of PANs are male. Male nurses tend to respect the physician-nurse hierarchy, avoid conflicts with doctors, and strive to build good relationships through personal interactions [29]. Physicians often prefer nurses who prioritize harmony and do not challenge the physician-nurse inequality, as they are seen as less threatening to the doctors' status. However, maintaining this hierarchy may hinder nurses from fully utilizing their unique abilities and impede efficient interprofessional collaboration [30]. Moreover, this hierarchy tends to result in physicians rating their own satisfaction higher than that of nurses when evaluating mutual satisfaction [31]. Therefore, it is crucial to continue monitoring whether an environment can be created where PANs and anesthesiologists are both satisfied with their professional relationships. In contrast, PANs can fully exercise their unique capabilities.

Anesthesiologists expressed concerns and criticisms about task shifting anesthesia services to PANs, whether for or against it. "Differences in patient perceptions" contributed to these concerns and criticisms. Traditionally, the status of nurses is considered lower than that of doctors in Japan [32]. Additionally, there are currently very few nurses with special skills, such as nurse practitioners, and patients are unfamiliar with the idea that "nurses are proactively involved in the medical treatment process" [33]. Therefore, anesthesiologists were concerned that the very fact that "only nurses are monitoring the patient while the patient is asleep under anesthesia" might offend the patient's feelings. Anesthesiologists also anticipated that more than merely explaining the excellence of PAN individuals would be required to change this patient sentiment. This anticipation led to requests for the national certification of PANs and recommendations for promoting the benefits and safety of the PAN system and PAN anesthesia to the public.

This study demonstrated significant variations in how anesthesiologists perceive the role of PANs. In particular, divergent perspectives emerged regarding PAN anesthesia and PANs as substitutes for anesthesiologists. Such differences in perspectives and conflicts may hinder effective task shifting. When nurses with specialized skills cannot find consistency in their required duties and feel ambiguous about their roles, their job satisfaction significantly reduces [34]. Low satisfaction is one of the risk factors for burnout syndrome among nurses. Moreover, differences in role perception between physicians and anesthesia nurses may lead to conflicts between the two groups. It has been noted that persistent conflicts not only cause considerable stress for both doctors and nurses but may also result in a decline in the quality of patient care [35].

It is important to clearly define the scope of practice and roles of PANs. However, when clarifying these roles, it is essential to incorporate opinions of both anesthesiologists and PANs. It has been noted that if role standardization proceeds without considering the needs

and expectations of nurses, there is a risk that nurses with higher levels of specialization may experience increased stress due to perceived gaps between hospital demands and their own roles [36]. It is necessary to clarify what both anesthesiologists and PANs seek, reconcile their respective desires, and explore better forms of collaboration.

The relationship between physicians and nurses is known to have various impacts on task shifting from physicians to nurses. To achieve successful task shifting, several key factors have been identified as essential: establishing a trusting relationship between physicians and nurses, clarifying the extent of authority delegation and responsibility sharing with nurses, and determining who bears the ultimate responsibility [10]. The detailed analysis of Japanese anesthesiologists' perceptions of PANs presented in this study suggests the need for a strategy that considers Japan's unique cultural and social background in addition to existing factors. This insight demonstrates the importance of considering universal challenges and culture-specific factors in implementing task shifting, providing a valuable perspective for healthcare policymakers and researchers in Japan and internationally.

Furthermore, the importance of legal frameworks for successful task shifting has been emphasized [10]. However, the PAN system in Japan was initiated without such legal development. As highlighted in this study, the diverse perspectives on the task shift from anesthesiologists to PANs underscore the potential for various confusions and conflicts of opinions arising from introducing the system without legal development. To foster more effective collaborative relationships and make meaningful contributions to Japanese society, it is essential to have comprehensive social discussions on task shifting to PANs, actively involving academic societies, nursing associations, and the government. Specifically, this could include establishing educational, certification, and practice standards as required by The International Federation of Nurse Anesthetists, as well as developing appropriate legal frameworks [6]. Until these milestones are achieved, the confusion and conflict in anesthesiologists' perspectives on this system may persist, potentially hindering effective task shifting. By addressing these challenges, we can expect the effective development of the PAN system in Japan and an improvement in the quality of healthcare. Moreover, Japan's experience could be a valuable lesson for other countries, particularly those where the legal status of nurses with new specialties is not clearly defined.

## Limitation

Firstly, a single researcher (MT) conducted all 24 interviews in this investigation, presenting a limitation and a strength of our study. This approach ensured a consistent standard of interview quality throughout the inquiry. As MT, an anesthesiologist aligning with the interviewees, who belonged to the intermediate generation of all participants and possessed experience acquiring industrial counseling techniques, she executed interviews with profound empathy, skillfully extracting diverse perspectives. Nonetheless, this methodology carried the potential to introduce bias into the outcomes. To mitigate the potential influence of MT's personal experiences and perspectives on data collection and interpretation, the research team engaged in iterative discussions concerning data analysis and sought guidance from SK, a seasoned qualitative research professional, thereby enhancing the validity of the interpretations.

Secondly, this study was limited by the participation of less experienced anesthesiologists. Among the 24 participants, three were senior residents, each with at least two years of experience in anesthesiology. Our analysis may not adequately reflect the perspectives of less experienced anesthesiologists, particularly those in their early years of practice. Previous research has identified specific challenges in teamwork between junior doctors and nurses, such as leadership issues due to inexperience and collaboration difficulties stemming from a lack of trust

and respect [37]. These challenges may also exist in the relationship between senior residents and PANs. Future research, including a more diverse range of experience levels, may provide additional insights into the relationships between anesthesiologists and PANs.

## Conclusion

This study is the first to reveal that Japanese anesthesiologists hold diverse perspectives on PANs and the system. The qualitative approach was well-suited for exploring their diverse perspectives. In particular, it was shown that perspectives on the role of PANs vary significantly among institutions and individuals, with some critical perspectives. The need for task-shifting to nurses has emerged as a strategic imperative to improve healthcare services and address rising medical costs. Our data provided crucial insights, including findings suggesting potential barriers to shifting anesthesia duties to PANs.

## Supporting information

**S1 Appendix. Interview guide.**
(DOCX)

**S2 Appendix. Final analytical framework.**
(DOCX)

**S3 Appendix. Application of analytical framework to verbatim records.**
(PDF)

**S4 Appendix. Standards for Reporting Qualitative Research (SRQR).**
(DOCX)

**S5 Appendix. A comprehensive overview of the coding structure.**
(PDF)

## Acknowledgments

We sincerely thank the 24 anesthesiologists for their invaluable contributions to this study. Their insights have significantly enriched our research.

## Author Contributions

**Conceptualization:** Mikiko Tamai, Shogo Kojima, Yasuko Baba, Kiyoyasu Kurahashi.

**Data curation:** Mikiko Tamai, Shogo Kojima, Yasuko Baba, Kiyoyasu Kurahashi.

**Formal analysis:** Mikiko Tamai, Shogo Kojima, Yasuko Baba, Kiyoyasu Kurahashi.

**Investigation:** Mikiko Tamai, Shogo Kojima.

**Methodology:** Mikiko Tamai, Shogo Kojima, Kiyoyasu Kurahashi.

**Project administration:** Mikiko Tamai, Shogo Kojima, Yasuko Baba, Kiyoyasu Kurahashi.

**Resources:** Mikiko Tamai.

**Software:** Mikiko Tamai, Shogo Kojima, Kiyoyasu Kurahashi.

**Supervision:** Shogo Kojima, Yasuko Baba, Kiyoyasu Kurahashi.

**Validation:** Mikiko Tamai, Shogo Kojima, Yasuko Baba, Kiyoyasu Kurahashi.

**Visualization:** Mikiko Tamai, Shogo Kojima, Yasuko Baba, Kiyoyasu Kurahashi.

**Writing – original draft:** Mikiko Tamai, Shogo Kojima, Yasuko Baba, Kiyoyasu Kurahashi.

**Writing – review & editing:** Mikiko Tamai, Shogo Kojima, Yasuko Baba, Kiyoyasu Kurahashi.

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
