## [Decision Letter · Decision Letter 0]

22 May 2024

PONE-D-24-11821Amid the chaos: a qualitative study of anesthesiologists' perspectives on Japanese perianesthesia nursesPLOS ONE

Dear Dr. TAMAI,

Thank you for submitting your manuscript to PLOS ONE. After careful consideration, we feel that it has merit but does not fully meet PLOS ONE’s publication criteria as it currently stands. Therefore, we invite you to submit a revised version of the manuscript that addresses the points raised during the review process.

We look forward to receiving your revised manuscript.

Kind regards,

Stefano Turi

Academic Editor

PLOS ONE

2. In the online submission form you indicate that your data is not available for proprietary reasons and have provided a contact point for accessing this data. Please note that your current contact point is a co-author on this manuscript. According to our Data Policy, the contact point must not be an author on the manuscript and must be an institutional contact, ideally not an individual. Please revise your data statement to a non-author institutional point of contact, such as a data access or ethics committee, and send this to us via return email. Please also include contact information for the third party organization, and please include the full citation of where the data can be found.

Reviewers' comments:

Reviewer's Responses to Questions

**Comments to the Author**

1. Is the manuscript technically sound, and do the data support the conclusions?

Reviewer #1: Yes

Reviewer #2: Partly

2. Has the statistical analysis been performed appropriately and rigorously? 

Reviewer #1: N/A

Reviewer #2: N/A

3. Have the authors made all data underlying the findings in their manuscript fully available?

Reviewer #1: Yes

Reviewer #2: Yes

4. Is the manuscript presented in an intelligible fashion and written in standard English?

Reviewer #1: Yes

Reviewer #2: Yes

5. Review Comments to the Author

Reviewer #1: Too long article that can be shorter by modifying extra data. Ref can be improved and more updated and needs more clarification in conclusion part. it would be better if the authors try to explain about the usefulness of the study findings for international audiences.

Reviewer #2: First, I congratulate the authors for choosing the topic of advanced nurse anesthetist practice, from the perspective of doctors and not nurses.

The study investigated the perception of anesthesiologists regarding the role of the nurse anesthetist in advancing practice in the Japanese healthcare system. The nurse anesthetist system was implemented in Japan in 2010, 14 years later, the authors sought to understand the perception based on the experience of anesthetists who play the role of supervisor of this advanced practice nurse. The abstract explains the content of the manuscript and encourages the reader to read the full text. The advanced practice nurse (APN) is a way of inserting this professional into health services, especially to cover the shortage of doctors in general specialties, such as anesthesiology. The problem is well founded, with current bibliographic references on the topic. The method is appropriate and responds to the objective of the study, by choosing the interview (in-depth) as a way of obtaining qualitative data. The participant eligibility criteria led to obtaining a representative sample of the set of qualified narratives from the professionals who volunteered to participate in the interviews. The ethical issues of the field research were presented with clarity and sufficient detail to understand the sensitive nature of the topic, while preserving confidentiality. The interview's trigger question - "Could you please outline the primary responsibilities of perianesthesia nurses in your hospital?" – elicited narratives that allowed an in-depth analysis. It also used a questionnaire with a set of open-ended questions. The analysis procedures were anchored in the framework method”, which are well described and encourage the reader to understand how the categories were established.

Despite this, there are some points in the manuscript that could be better presented to ensure the scientific criteria of qualitative studies.

6. PLOS authors have the option to publish the peer review history of their article (what does this mean?). If published, this will include your full peer review and any attached files.

Reviewer #1: No

Reviewer #2: **Yes: **Ivone Evangelista Cabral, RN, PhD. Associate Professor and Researcher. Universidade do Estado do Rio de Janeiro. Rio de Janeiro, Brasil

---

## [Author Response · Author response to Decision Letter 0]

2 Aug 2024

Dear Reviewers,

We sincerely thank you for your valuable comments and suggestions, which have helped us improve the quality of our manuscript. We have carefully addressed each point raised by the reviewers. 

Critical revisions include changes to our discussion and conclusion sections and a new appendix with our coding structure. We expanded our discussion to contextualize these findings within the broader framework of advanced nursing practice.

Due to the detailed nature of our responses, we have prepared a comprehensive document titled "Response to Reviewers", which we have uploaded as a separate file. This document contains our point-by-point responses to all comments and suggestions, along with detailed explanations of the changes made to the manuscript.

We kindly ask you to refer to this document for a full understanding of our revisions and responses. We believe that these revisions have significantly improved the manuscript and hope they adequately address your concerns.

We look forward to your further review and feedback.

Sincerely,

Mikiko Tamai

---

## [Decision Letter · Decision Letter 1]

9 Sep 2024

PONE-D-24-11821R1Variabilities and contentions in anesthesiologists' perspectives on Japanese perianesthesia nurses: a qualitative studyPLOS ONE

Dear Dr. TAMAI,

Thank you for submitting your manuscript to PLOS ONE. After careful consideration, we feel that it has merit but does not fully meet PLOS ONE’s publication criteria as it currently stands. Therefore, we invite you to submit a revised version of the manuscript that addresses the points raised during the review process.

Please submit your revised manuscript by Oct 24 2024 11:59PM. If you will need more time than this to complete your revisions, please reply to this message or contact the journal office at plosone@plos.org. Please include the following items when submitting your revised manuscript:A rebuttal letter that responds to each point raised by the academic editor and reviewer(s). You should upload this letter as a separate file labeled 'Response to Reviewers'.A marked-up copy of your manuscript that highlights changes made to the original version. You should upload this as a separate file labeled 'Revised Manuscript with Track Changes'.An unmarked version of your revised paper without tracked changes. You should upload this as a separate file labeled 'Manuscript'.If applicable, we recommend that you deposit your laboratory protocols in protocols.io to enhance the reproducibility of your results. Protocols.io assigns your protocol its own identifier (DOI) so that it can be cited independently in the future. For instructions see: https://journals.plos.org/plosone/s/submission-guidelines#loc-laboratory-protocols. Additionally, PLOS ONE offers an option for publishing peer-reviewed Lab Protocol articles, which describe protocols hosted on protocols.io. Read more information on sharing protocols at https://plos.org/protocols?utm_medium=editorial-email&utm_source=authorletters&utm_campaign=protocols.

We look forward to receiving your revised manuscript.

Kind regards,

Stefano Turi

Academic Editor

PLOS ONE

Journal Requirements:

**Additional Editor Comments **

Dear Authors thank you for submitting your revised version of the manuscript. However, in my opinion, I think that you did not completely follow the indications of Reviewer 1 "Too long article that can be shorter by modifying extra data". I suggest to shorten the article and to present differently some parts in order to make your work more easily interpretable.

Thank you

Stefano Turi 

Reviewers' comments:

Reviewer's Responses to Questions

**Comments to the Author**

1. If the authors have adequately addressed your comments raised in a previous round of review and you feel that this manuscript is now acceptable for publication, you may indicate that here to bypass the “Comments to the Author” section, enter your conflict of interest statement in the “Confidential to Editor” section, and submit your "Accept" recommendation.

Reviewer #2: All comments have been addressed

2. Is the manuscript technically sound, and do the data support the conclusions?

Reviewer #2: Yes

3. Has the statistical analysis been performed appropriately and rigorously? 

Reviewer #2: N/A

4. Have the authors made all data underlying the findings in their manuscript fully available?

Reviewer #2: Yes

5. Is the manuscript presented in an intelligible fashion and written in standard English?

Reviewer #2: Yes

6. Review Comments to the Author

Reviewer #2: The manuscript brings to light the growing role of advanced practice nurses in anesthesia in Japan. The study aims to uncover anesthesiologists' perspectives on PANs and the PAN system. The authors have meticulously incorporated the recommendations from the initial manuscript evaluation and adhered to the SQUIRE checklist to report the qualitative research results. The method description provides a comprehensive understanding of the study's implementation. The results were effectively communicated using narratives, tables, and figures. The discussion distills new knowledge into five distinct categories: Anesthesiologists' perspectives on the implementation of the PAN system in Japan; High appraisal of current PANs Anesthesiologists' expectations; Anesthesiologists' perspectives on PAN anesthesia; and Anesthesiologists' perspectives on the PAN system. Given these points, it is my belief that the manuscript possesses scientific merit worthy of publication since was applied rigor and all the steps of the study's implementation.

7. PLOS authors have the option to publish the peer review history of their article (what does this mean?). If published, this will include your full peer review and any attached files.

Reviewer #2: **Yes: **Ivone Evangelista Cabral, PhD Nurse

Adjunct Professor

State University of Rio de Janeiro, Brasil

---

## [Author Response · Author response to Decision Letter 1]

3 Oct 2024

Dear Dr. Stefano Turi,

Thank you for your valuable feedback on our manuscript. We have carefully addressed your comments regarding the need to shorten the article and improve its interpretability. Specifically:

1. We have significantly reduced the text volume in the Methods section by presenting information visually through figures and tables.

2. We have condensed data in the Results section without compromising meaning, thereby reducing the overall word count.

3. We have thoroughly reviewed and updated all references, including publication details and access dates where necessary.

These changes have resulted in a more concise and easily interpretable manuscript while maintaining its scientific integrity. We believe these revisions adequately address the concerns raised by Reviewer 1 regarding the article's length.

Response to Journal Requirements:

We have carefully reviewed our reference list to ensure it is complete and correct. No retracted papers have been cited in our manuscript. All references have been updated with the most current information available.

Response to Reviewer #2:

We sincerely appreciate your positive feedback and your recommendation for publication. We are grateful for your recognition of our efforts to incorporate previous recommendations and adhere to the SRQR checklist. We have maintained the scientific rigor and comprehensive reporting of our study throughout the revision process.

We hope these revisions meet your expectations and those of PLOS ONE. We look forward to your feedback.

Sincerely,

Mikiko Tamai

---

## [Editor Report · Decision Letter 2]

21 Oct 2024

Variabilities and contentions in anesthesiologists' perspectives on Japanese perianesthesia nurses: a qualitative study

PONE-D-24-11821R2

Dear Dr. Mikiko Tamai,

We’re pleased to inform you that your manuscript has been judged scientifically suitable for publication and will be formally accepted for publication once it meets all outstanding technical requirements.

Kind regards,

Stefano Turi

Academic Editor

PLOS ONE

---

## [Editor Report · Acceptance letter]

24 Oct 2024

PONE-D-24-11821R2 

PLOS ONE

Dear Dr. TAMAI, 

I'm pleased to inform you that your manuscript has been deemed suitable for publication in PLOS ONE. Congratulations! Your manuscript is now being handed over to our production team.

Kind regards, 

on behalf of

Dr. Stefano Turi 

Academic Editor

PLOS ONE